# Composite Nanomaterials Based on Polymethylmethacrylate Doped with Carbon Nanotubes and Nanoparticles: A Review

**DOI:** 10.3390/polym16091242

**Published:** 2024-04-29

**Authors:** Lusine Elbakyan, Irina Zaporotskova

**Affiliations:** Institute of Priority Technologies, Volgograd State University, 100 Prospect Universitetsky, Volgograd 400062, Russia; irinazaporotskova@gmail.com

**Keywords:** poly(methyl methacrylate), carbon nanotubes, polymer nanocomposite, nanoparticles, adsorption interaction, electronic energy structure, mechanical properties, DFT calculations

## Abstract

Composite polymer materials have high strength and lightness, which makes them attractive for use in a variety of structures and products. The present article contains an overview of modern works devoted to the production of composite materials based on poly(methyl methacrylate) (PMMA) with improved characteristics. The possibility of obtaining such materials can be a key area for creating more efficient and durable products in various industries. Various methods were considered to improve the characteristics of PMMA by doping the polymer matrix with carbon nanotubes (CNTs), graphite, nanohydroxyapatite particles, micro-zirconia nanoparticles, titanium dioxide, etc. The possibilities of using the obtained composite materials in various industries such as aviation, automotive, construction, medical and others are discussed. This article also presents the results of our own research on the mechanisms of interaction of PMMA with single-layer CNTs, leading to the creation of a composite polymer system “PMMA+CNT”, achieved using the modern quantum chemical method DFT. This article presents a review of the recent research on the effect of CNTs on the mechanical and electrically conductive properties of nanocomposite materials. The outcomes of this study can be important for the development of science and technology in various fields, from fundamental chemistry to applied scientific research.

## 1. Introduction

Nanotechnology and industrial nanomaterials are a growing industry that creates many economic and social benefits [1,2,3]. Therefore, the development of new nanoscale materials is one of the most important tasks of our time. The analysis of the development of promising materials and technologies shows that currently, the main efforts of researchers are focused on the creation of new nanoscale materials, as well as on the study of their unique properties and potential applications [4,5,6,7]. Thus, scientists around the world are working on developing methods for producing nanoparticles of various shapes and sizes, exploring their electrical, optical, magnetic and mechanical properties. In addition, researchers are striving to create new composite materials reinforced with nanoscale particles with improved characteristics such as strength, thermal conductivity, electrical conductivity and others that can be useful in various fields of science and technology [8,9]. Interest in the creation of nanostructures is also associated with the search for new methods of producing composite materials based on known studied substances which can be used in industry, medicine, energy and other industries.

The creation of polymer nanocomposites is receiving much attention. A polymer nanocomposite is a two-phase material consisting of a polymer matrix and embedded nanoparticles of another material. The polymer matrix forms the main phase, in which nanoparticles are embedded, forming the second phase. This two-phase device provides unique properties and characteristics of a composite material that differ from those of a pure polymer or embedded nanoparticles individually. 

Carbon-containing nanostructures can be used as active components of a nanocomposite, thanks to which materials can be created that serve as both optical media and semiconductors [10].

Despite a fairly large number of works devoted to the study of the structure and properties of nanostructured composites, there are still many problems that require solutions, since the systems that can be called nanocomposites are very diverse. Currently, there are a large number of papers devoted to the study of polymer nanocomposites, and various polymer matrices and various fillers of these matrices are considered [9,10,11,12,13,14,15]. Among these polymers is the polymer material polymethyl methacrylate (PMMA), known for its high impact strength, light transmittance and biocompatibility [16]. Polymethylmethacrylate is used in lighting engineering, medicine, aviation, mechanical engineering, etc. However, the thermal stability and mechanical properties of PMMA are not good enough to meet industry demands. Therefore, it is an urgent issue to make various changes to the polymer to improve its physical properties. Quite a significant number of modern studies have been devoted to the study of the possibility of modifying PMMA.

This article provides an overview of scientific works in recent years devoted to the creation of polymer nanocomposites based on polymethylmethacrylate doped with CNTs, graphite nanoparticles, nanohydroxyapatite, zirconium dioxide nanoparticles, titanium dioxide, etc., and discusses the possibilities of using the obtained composite materials in various industries such as aviation, automotive, construction and medical. The current article also presents the results of our own research on the mechanisms of interaction of PMMA with single-layer CNTs, leading to the creation of a composite polymer system “PMMA+CNT”, achieved using the modern quantum chemical method DFT [17]. The results of our own experimental studies on the effect of the number of CNTs on the mechanical and electrically conductive properties of the composite are presented. The study and analysis of recent works devoted to the modification of the known polymer material polymethylmethacrylate may be important for the development of science and technology in various fields, from fundamental chemistry to applied scientific research using modern composite materials with improved properties.

## 2. Polymethylmethacrylate

Polymethylmethacrylate is a lightweight synthetic polymer with good physical, mechanical and electrical insulating characteristics. PMMA is not a toxic material, which significantly expands the scope of application of this material. Obtaining new modified materials based on it with pre-predicted physical and mechanical properties will expand the scope of application of polymers based on PMMA [18].

The use of PMMA as an electron-blocking layer in diodes with quantum dots (QDs) is described in [19,20]. So, to obtain such layers, the polymer has been previously dissolved in acetone in [20]. The concentration of PMMA in the solvent ranged from 0.05 to 1.2 mg/mL. Quantum dots have a “core/multilayer shell” structure. To improve the brightness characteristics and output current, an electron-blocking layer of polymethyl methacrylate was added to the QD structure. The study showed that during layer-by-layer deposition, the concentration of the PMMA solution plays a significant role in obtaining highly efficient QD-based LEDs (QDLEDs). In particular, at a high concentration of 1.2 mg/mL, the current and brightness characteristics of the QDLED were reduced compared to a similar device without an electronic-blocking layer (EBL). At a low concentration of the PMMA solution, the characteristics of QDLEDs (brightness, output current, and reduction in switching voltage) were improved. The best output current parameter was the QDL sample with the EBL deposited from a PMMA solution with a concentration of 0.4 mg/mL. In the case of a minimum concentration of 0.05 mg/mL of the PMMA solution, the brightness of the resulting LEDs was 18,671 cd/m^2^, which is four times higher than these values for the devices without a blocking layer.

PMMA plays an important role in medicine due to its unique properties. The biocompatibility of PMMA makes it a safe material for contact with body tissues [21,22]. PMMA acrylic resin is widely used in dentistry as a material for dentures [23,24,25,26]. Its use is related to its biological, physical and esthetic characteristics. However, some of its disadvantages, such as shrinkage during polymerization, water absorption and low mechanical properties, have led to thoughts about the need to modify acrylic resins. And one of the main ways to modify polymethylmethacrylate is the doping of the PMMA polymer matrix with various fillers.

### 2.1. Doping of PMMA with Various Nanoparticles

Currently, various nanofillers are used to improve the properties of the PMMA matrix, including graphene nanofillers, nanoparticles of titanium dioxide, zirconium dioxide, nanohydroxyapatite, etc. [27,28]. Thus, graphene nanoplatelets were used as nanoadditives in the PMMA polymer matrix in [27]. The first step was the process of dispersing graphene nanoplatelets in liquid PMMA monomers, and then, the polymerization process was carried out. The study showed that the resulting composite material had excellent mechanical properties, electrical conductivity and thermal stability, and a lower density. The authors considered a method for obtaining a composite material for dentistry based on PMMA, which was doped with titanium dioxide nanoparticles (PMMA/1 wt %TiO_2_ and PMMA/2 wt %TiO_2_) to obtain better characteristics. PMMA/TiO_2_ nanocomposites have been successfully synthesized by mixing the melt using a twin-screw extruder in [28]. It was found that the composites have traditional hardness, Martens hardness and scratch resistance higher than pure PMMA. This occurs due to the reinforcement of the polymer matrix with titanium oxide. The authors concluded that the obtained composites are promising for use in dentistry.

The most common types of nanomaterials used to improve the properties of polymer matrices based on PMMA are presented in [29]. Depending on the nature of the nanofillers used, the author identified 1—organic fillers (dendrimers, micelles, liposomes, polymer nanoparticles and ferritin); 2—inorganic fillers (metal nanoparticles Ag, Au, Cu, metal-oxide nanoparticles (for example, Fe_3_O_4_, ZnO, MgO and TiO_2_), hydroxyapatite and mesoporous silica); 3—carbon-based nanofillers (fullerenes, quantum dots, carbon nanotubes, graphene and nanodiamonds). Next, we will consider in more detail the works devoted to the creation of nanocomposites based on polymethylmethacrylate with various nanofillers.

There are many studies that have proven that the addition of graphite nanoparticles, nanohydroxyapatite, zirconium dioxide, titanium dioxide, and carbon nanotubes does not lead to biocompatibility problems and does improve some properties of PMMA, namely, mechanical, tribological, thermal and antimicrobial properties, which is presented in [30,31,32,33,34].

The effect of SiO_2_ (silicon oxide) nanoparticles on the bending strength of the acrylic matrix used in dentures, depending on the concentration of the SiO_2_ nanoparticles (0.25, 0.5 and 0.75 wt %), was described in [35]. It was found that the best result (maximum bending strength) is achieved with a SiO_2_ content of 0.25% by weight. To improve the mechanical properties of dentures based on PMMA, [36] proposed a method of strengthening the material with graphene–silver (Gr-Ag) nanoparticles. Samples of the polymer composite containing 1 wt % and 2 wt % Gr-Ag nanoparticles were tested for compressive, flexural, tensile, water absorption and morphology strengths. It was found that 1% nanoparticle content is sufficient for the material to withstand higher loads and exhibit higher flexural and tensile strength compared to unmodified PMMA. A decrease in water absorption due to a decrease in porosity was also found. All of this can be used to create a compact and durable design.

The study of the characteristics of composites made with graphite reinforcements that are based on polymethylmethacrylate is focused on in [37]. Injection molding was used to produce such composites. The authors presented experimental studies of the strength characteristics and impact strength of a composite material based on PMMA with a different percentage of graphite. Thus, at 85% PMMA and 15% graphite, the impact strength was improved to 5.33 kJ/m^2^; at 90% PMMA and 10% graphite, the impact strength was 7.27 kJ/m^2^. The optimal hardness value was obtained with 85% PMMA and 15% graphite.

### 2.2. Carbon Nanotubes as Materials for Doping into PMMA

One of the most popular materials used as nanoadditives for polymer materials is carbon nanotubes (CNTs), which have unique physico-chemical properties. Nanotubes are structures consisting of one or more hexagonal graphene layers rolled into tubes. Their surface consists of regular hexagonal carbon cycles (hexagons). The diameter of the nanotubes is from one to several nanometers, and the length is up to several microns [38,39,40,41,42].

The electrical properties of carbon nanotubes depend on the type of rolling of the hexagonal layers. This can affect whether the nanotube functions as a conductor or a semiconductor. The electrical resistance of carbon nanotubes can change when other molecules attach to them.

Chirality is one of the most important properties of carbon nanotubes, which determines its structural property. The chirality of CNTs is the mutual orientation of the hexagonal grid and the longitudinal axis of the nanotube. The chirality of a nanotube is described by two integers (n and m), which determine the way in which the surface is twisted. The change (n and m) makes it possible to obtain SWCNTs of different diameters and chirality (Figure 1) [43]. There are two types of such nanotubes: chiral (SWCNTs with helical symmetry) and achiral (SWCNTs with cylindrical symmetry). Regarding the rolling of the layers, achiral SWCNTs have (1) “zig-zag” tubes—two edges of each hexagon are parallel to the axis of the cylinder; and (2) “arm-chair” tubes—two edges of each hexagon are perpendicular to the axis of the cylinder. The electrical properties of CNTs (metallic or semiconducting) depend on the type of chirality of the CNTs [42].

Due to their extraordinary properties, carbon nanotubes have been able to revolutionize an array of industries as a result of materials science and composite materials research [44,45]. The unique mechanical characteristics of CNTs have made it possible to use them in a matrix of polymer composites as a reinforcing filler. There are a huge number of studies devoted to the study of the properties of CNTs, as well as the possibility of using CNTs in polymers as additives in order to create new advanced classes of materials [46,47,48,49,50,51,52].

Currently, the problem of creating electrically conductive polymer materials is becoming more and more urgent. Such composites can be obtained by including electrically conductive fillers in their composition. Many studies have been devoted to the possibility of using carbon nanotubes as fillers of polymer matrices to create electrically conductive composite materials [53,54]. As it is known, CNTs have unique indicators of electrical and thermal conductivity. Also, the high aspect ratio of CNTs reduces the resistance of thermal contact between CNTs and the polymer matrix. This makes it possible to demonstrate the excellent electrical conductivity and thermal characteristics of composite materials. Thus, the use of carbon nanotubes as an electrically conductive filler will allow us to obtain a new class of polymer materials with conductive properties. Such composites can be widely used in various fields of science and technology.

There are also studies on the possibility of using polymer materials with incorporated CNTs in medicine. Thus, research on the development of nanocomposite materials for dental applications is presented in [33,55,56,57]. Special attention is paid to issues related to the regeneration of dental bone when using various types of nanocomposite materials.

It is important to know that the mechanical and other properties of any polymer nanocomposite depend on the method of dispersion, the quality of the materials and the degree of uniformity of the distribution of nanotubes in the polymer matrix. One of the important tasks in obtaining CNT-reinforced composite materials is the uniform dispersion of CNTs in a polymer matrix.

Many attempts have been made to incorporate CNTs into various polymer matrices, including polymethylmethacrylate. For example, as the most effective method for obtaining composite materials based on PMMA doped with multiwalled CNTs (PMMA/MWCNT), it was proposed to disperse CNTs in a liquid monomer using ultrasonic mixing before introducing carbon nanotubes into a polymer powder [58,59]. This method has made it possible to achieve significant improvements in both the static and fatigue properties of the obtained materials.

MWCNTs were added to the polymer made of PMMA, which is used as the basis of dentures [33]. The content of multilayer carbon nanotubes in the samples was 0.5 wt %, 1 wt % and 2 wt %. Then, the liquid polymer was stirred by using ultrasound for 20 min. Raman spectroscopy has shown that an interphase reaction occurs between the MWCNTs and the PMMA matrix. The results of static and dynamic studies have shown that the interfacial coupling between the MNTs and the PMMA is weak and needs to be improved. The use of MNTs up to 1% improves the strength and elasticity when bending the PMMA polymer. A larger amount of MWCNTs (2%) negatively affects the fatigue resistance of the MWCNT-PMMA complex. The author points out that this may be due to the poor dispersion of MWCNTs. The paper indicates that the only drawback of the obtained composites is the high cost and color (the material has a black color). Therefore, the authors suggest using such materials in the area of the midline of a complete denture, since this is an inconspicuous place.

In [60], CNTs are considered a possible method of delivery of various pharmaceutical components. Drug delivery systems are typically designed to improve the pharmacological activity and therapeutic effects of specific therapeutic agents. In that study, CNTs were modified to transmit several drugs with the ability to recognize optical signals for visualization in the treatment of cancer and some other infectious diseases.

### 2.3. The Use of CNTs to Improve the Strength Characteristics of PMMA

One of the important ways to create polymer composites with high application potential is the method of reinforcing polymers with carbon nanotubes. This leads to an increase in the elastic modulus and compressive strength in such composites. There is a fairly large number of studies focused on the assessment of the strength, thermomechanical, electrical and other physico-mechanical characteristics of polymer nanocomposites based on PMMA modified with introduced CNTs [61,62,63]. Composites reinforced with MWCNTs (2, 5 and 10 wt %) were prepared. Mechanical tests have shown that with an increase in the amount of MWCNTs, the elastic modulus and hardness of the composite increase compared to pure polymer [61]. In [62], PMMA/MWCNT composites were prepared by simply mixing a solution of PMMA with MWCNTs at various concentrations (2.5 wt %, 5 wt % and 10 wt %). It was found that the elastic modulus and hardness of the PMMA/MWCNT composite with a nanotube content of 5 wt % increased by 44% and 27% compared to pure PMMA. However, for composites containing 10 wt % nanotubes, the hardness and elastic modulus decreased by 28% and 23% compared to the composite containing 5 wt % MWCNTs. Dynamic mechanical analysis (DMA) of the PMMA/MWCNT composites was carried out. The complex elastic modulus for a 5 wt % PMMA/MWCNT composite and a pure PMMA sample was compared in dynamic mode for the frequency range of 50–210 Hz. It was found that the elastic modulus of the composite containing 5 wt % nanotubes increased over the entire frequency range. Thus, the advantage of compositions with a low content of MWCNTs in PMMA for applications in the range of low- and medium-load frequencies was proven in [62].

However, this problem is still relevant today, as the search for the optimal number of nanotubes continues, leading to the creation of necessary materials with specified characteristics and the search for new technologies for producing such composites. The properties of PMMA-based composite materials doped with CNTs were investigated in [64]. It is established that the PMMA/CNT mixture creates a cement that combines the properties of a multifunctional polymer with the mechanical reinforcing properties of CNTs. Figure 2 shows a schematic representation of a composite material based on polymethylmethacrylate doped with carbon nanotubes. The encouraging results of this work indicate the emergence of a new highly efficient elastic borehole cement suitable for drilling oil and gas wells.

Composites were obtained using hot pressing and injection molding methods. The conducted studies [65] showed that an increase in the concentration of CNTs in polymethylmethacrylate/CNT composites leads to an increase in the modulus of elasticity and reduces the deformation of the fracture of composites. Samples manufactured by two methods were studied: hot pressing (HP) and injection molding (IM). Samples of pure PMMA were tested, as well as PMMA containing CNTs at concentrations of 0.5 wt %, 1 wt %, 2 wt % and 6 wt %. For samples made by injection molding, samples with nanotube contents of 1, 2, 4, 5.5 and 8.5% were tested.

In [66], the dependence of the mechanical properties of PMMA/CNT composites produced by hot pressing and injection molding on processing parameters, including pressure, holding time, melt injection temperature and CNT concentration, was studied. It was found that tensile strength and elongation at failure decreased by approximately 30 and 40%, and hardness increased slightly with increasing CNT concentration from 0 to 1.5%.

In medicine, acrylic bone cements are used as a load transfer material between bone and a prosthetic implant during joint replacement surgery. This material has many advantages: a fast polymerization reaction, ease of preparation and ease of use. However, due to insufficient mechanical properties and insufficient biological activity, the use of PMMA is not effective enough. Acrylic bone cements do not adhere directly to the bone. Indirect surface adhesion increases the likelihood of a gap between the bone and the cement and between the cement and the implant [67,68]. This can cause the implant to become loose and bacteria to spread. Doping of PMMA with carbon nanotubes was proposed to produce biologically active cement in [69]. The addition of graphene oxide rGO at a concentration of 0.5 wt % increased the compressive modulus and strength of PMMA/CNT/rGO composite cements to 187.48 ± 5.79 MPa and 2058.50 ± 39.61 MPa, respectively. That is, graphene oxide further improves the mechanical properties of cements compared to the addition of carbon nanotubes alone. Thus, it seems that the new biologically active cement is a promising material for joint replacement operations.

### 2.4. The Use of PMMA/CNT Composite Polymer Materials to Protect against Electromagnetic Interference and Improve the Electrical Characteristics of the Source Material

Currently, due to the rapid growth of problems with electromagnetic interference (EMI), special attention is being paid to the materials that protect against electromagnetic radiation. Due to their characteristics such as being lightweight and anticorrosive and having high mechanical strength good chemical stability, carbon materials with a foamed structure demonstrate great potential in protection against electromagnetic interference and have attracted increasing attention in the last few years. The problem of developing new materials for electromagnetic shields with increased efficiency is very relevant, since electromagnetic resources are widely used, and the number of operating electronic devices are constantly increasing. The use of composite and carbon materials for protection against electromagnetic fields is a promising direction.

There are many works related to the introduction of carbon nanotubes into the PMMA matrix in order to develop new materials for electromagnetic shields with increased efficiency [70,71]. Thus, the characteristics of electromagnetic radiation absorption by a nanocomposite foam made of PMMA doped with carbon nanotubes with a laminated configuration were studied in [70]. It has been established that the absorption properties of electromagnetic waves are improved due to the layered structure of nanocomposite foams. The absorption band of laminated foam was 3.5 GHz (8.9–12.4 GHz). In nanocomposite foam with a thickness of 2 mm, this indicator is 0.8 GHz less. Similar results were obtained in a study on reflection losses in a wide frequency range (those of the nanocomposite foam are lower than those of the single-layer foam). The authors point out that this is due to the multilayer structure, which reduces the reflectivity of microwave radiation in the upper layer and, at the same time, reduces the loss of microwave radiation in the lower layer.

Various approaches to the synthesis of carbon materials with a foamed structure, including 3D printing, rapid solvent evaporation, supercritical foaming process, the carbonization method, etc., are described in [72]. The properties of electromagnetic interference shielding of various carbon materials with a foamed structure, including polymer-based composite materials reinforced with CNTs, are also discussed. Finally, the existing problems and possible solutions in this area are discussed. The excellent flexibility, thermal stability, mechanical strength, electrical conductivity, and sensory, capacitive and radiation-protective properties of PMMA-based nanocomposite polymers and nanofillers are suitable for many technical applications.

Special attention is paid to structural designs to improve the absorption of electromagnetic waves by polymer nanocomposites. Danfeng Zhou and other authors [73] proposed designs from alternately arranged sheets of pure PMMA and PMMA/CNT in order to obtain foam with an alternative gradient structure. Theoretically calculated and directly tested results show that compared with the normal single-layer foam, the absorbing bandwidth of the foam with AGS broadens to 3.2 GHz (8.8–12.0 GHz) and the minimum reflection loss decreases to −30.2 dB at the same CNT content. This provides an effective way to design and manufacture lightweight and high-quality materials that absorb electromagnetic radiation.

The possibility of forming composite granules of polymethylmethacrylate containing carbon nanotubes, which have good electrical conductivity and high optical transparency, is reported in [74]. Despite the use of a small number of nanotubes (0.68~6.8) × 10^−2^ vol.%, conductive channels were created due to the homogeneous decoration of CNTs on PMMA particles obtained by electrostatic assembly. The study investigated the electrical conductivity and transparency of manufactured CNT-PMMA composite tablets. The results of that study can play an important role in the creation of lightweight, conductive and transparent polymer composites with the necessary physical properties.

Another method for obtaining composite foam from PMMA and multiwalled CNTs for protection against electromagnetic interference is proposed in [75]. The claimed composite material was obtained using a solution coating process and super-critical liquid foaming technology. The resulting composite foam has good electrical conductivity, with a low filler content and a percolation threshold of 0.019 vol.%. This was obtained using the selective distribution of MWCNTs at the interface of the PMMA phases and the selective concentration of MWCNTs inside the cell walls. Also, composite foam has good electrical conductivity with a low filler content of microspheres, which are a three-layered microcellular structure formed from PMMA. Due to the isolated structure and multimodal cellular structure, the PMMA/MWCNT composite (5.0 wt %) has a fairly low density (0.49 g/cm^3^), while the electrical conductivity (3.19 S/m) and shielding efficiency (35.9 dB) are quite high. It was found that the specific shielding efficiency (356.5 dB·cm^2^/g) of the obtained material is higher than that of many known polymer nanocomposites. Nanocomposite foams also have good mechanical properties.

PMMA granules coated with MNTs were dispersed in a thermoplastic polyurethane (TPU) matrix through melt mixing in [76]. The effect of PMMA granule size on the pressure sensitivity was measured and compared. The results of the pressure sensitivity show that with an increase in the applied pressure on the composites, the PMMA granules coated with MNTs present in the TPU matrix moved closer to each other and formed three-dimensional conductive mesh structures at the threshold of seepage through the interface of PMMA coated with MNTs. The presence of granules of PMMA with MNTs contributed to achieving early contact even at low loads. The composites also had improved thermal conductivity. The authors recommended the obtained materials as ideal candidates for use in pressure measurement systems.

The possibility of developing MNT-doped composites that form a stable percolation grid and demonstrate high electrical conductivity is discussed in [77]. The halogenation process is described, which can provide effective control of the electrical characteristics of PMMA/MNT nanocomposites. In that article, the authors have shown the possibility of using a simple technology to produce composites with a spatial localized distribution of carbon nanotubes. The resulting composite material (MWCNT content: 1.55%) has a high electrical conductivity, which is 2.86 × 10^−2^ Cm/cm, and a low percolation threshold of about 0.37% vol.

The technology for producing flexible monopole antennas from polymethylmethacrylate filled with single-walled carbon nanotubes (CNTs) was proposed in [78]. Reflection coefficients, radiation patterns and maximum gain of non-bent and curved antennas from the SWCNTs were estimated. Based on the results obtained, a forecast was made about the possibility of using PMMA/SWCNT films for the manufacture of flexible antennas. Rectangular and round polymer antennas made of PMMA were presented by the authors in [79]. Instead of traditional copper, carbon nanotubes (CNTs) were used for the conductive part and the grounding plane. It was established that the resulting designs find a suitable compromise between size, gain and efficiency. This proves the possibility of their practical application in real conditions.

The reviewed articles proved the possibility of creating new materials with improved characteristics by doping the polymer matrix with carbon nanotubes. This emphasizes the importance of a low percentage of CNTs (from 0.5 to 1.5%).

## 3. Theoretical and Experimental Studies of Polymer Composite Materials Based on Polymethylmethacrylate and Carbon Nanotubes

The mechanisms of formation and some physico-chemical properties of composite structures based on CNTs doped with PMMA are investigated.

A computer simulation of the interaction of PMMA polymer fragments with the surface of single-layer CNTs of types (6, 6), (6, 0) and (7, 1) used to modify the polymer and create a new composite polymer material was carried out [80,81]. The choice of the types of single-layer CNTs is due to the fact that they belong to different types of chirality and have different physical properties. The research was carried out within the framework of the DFT calculation method [82,83]. The model of the PMMA/CNT system is shown in Figure 3.

The values of the interaction energies in all of the considered cases revealed the occurrence of a physical interaction of PMMA with the CNTs.

Thus, the established fact of the PMMA interaction with the surface of single-layer CNTs explains the mechanism of creating a composite polymer material based on polymethylmethacrylate reinforced with nanotubes, leading to the creation of stable complexes.

To conduct experimental studies and prepare samples of composites based on PMMA doped with CNTs, we used a powdered polymer PMMA. The polymerization process of the selected polymer occurs when a suspension liquid is added without additional conditions (at room temperature). Ultrasonic mixing technology is used for the most uniform dispersion of CNTs in a polymer matrix. The claimed technology, as the most effective method for obtaining PMMA/CNT composite materials, is described in detail in [80,81,83,84,85]. Some physico-mechanical characteristics of samples of a polymer composite material based on PMMA reinforced with CNTs at various concentrations, namely, 0.01 wt %, 0.03 wt % and 0.05 wt %, such as the maximum permissible load, destructive load and hardness, are assessed according to the Rockwell method. The established patterns of changes in the strength characteristics of the polymer composite PMMA/CNTs can form the basis for establishing the physical principles of creating new materials with improved strength characteristics, which will lead to a significant expansion of the use of polymer materials in various fields of industry, energy, electronics, medicine, etc.

The conductivity of a PMMA-based nanocomposite material with the addition of various amounts of CNBs has been studied. A non-linear dependence of the conductivity on the applied voltage has been found for various frequencies. The experimental dependencies obtained [83,84,85] make it possible to control the amplitude of a given harmonic using an alternating electric field in a wide range of values.

## 4. Conclusions

As shown by the analysis of works devoted to theoretical and experimental studies of polymer composites based on polymethylmethacrylate doped with nanoparticles and carbon nanotubes, the possible application of these nanocomposites is focused on the fields of materials science and nanotechnology. This material is the most promising material for further use, and the development of technologies for its production leads to the creation of new physical objects, the properties of which are of both scientific and applicative interest. Current research is focused on the search for new modifying additives that will improve the characteristics of the polymer material based on PMMA. This review highlights not only the unique physico-chemical properties of a polymer composite based on polymethylmethacrylate but also the possible synergistic effects that arise when modifying PMMA with various nanofillers—carbon nanotubes and nanoparticles. The new polymer composites created in this way have improved mechanical, electrical and radio-absorbing properties, which opens up prospects for their use in various industries (medicine, construction, electronics, aviation, automotive, etc.).

This study was carried out within the framework of the state task of the Ministry of Science and Higher Education of the Russian Federation (subject “FZUU-2023-0001”).

## Figures and Tables

**Figure 1 polymers-16-01242-f001:**
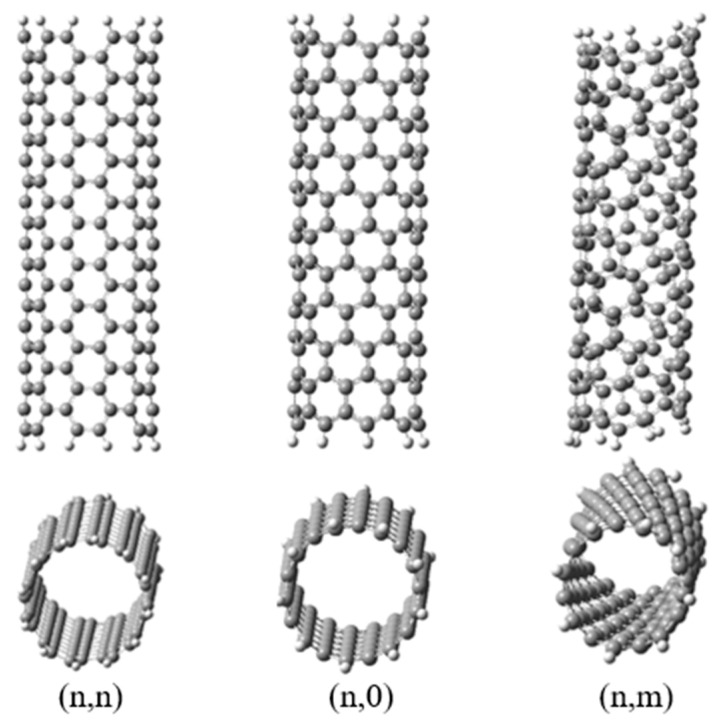
SWCNTs of different chirality.

**Figure 2 polymers-16-01242-f002:**
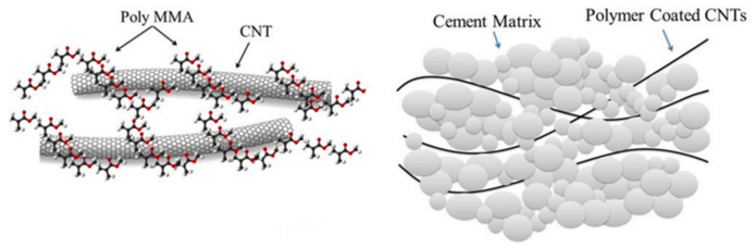
The mechanism of a reinforced cement sheath for the distribution of the surface stresses [64].

**Figure 3 polymers-16-01242-f003:**
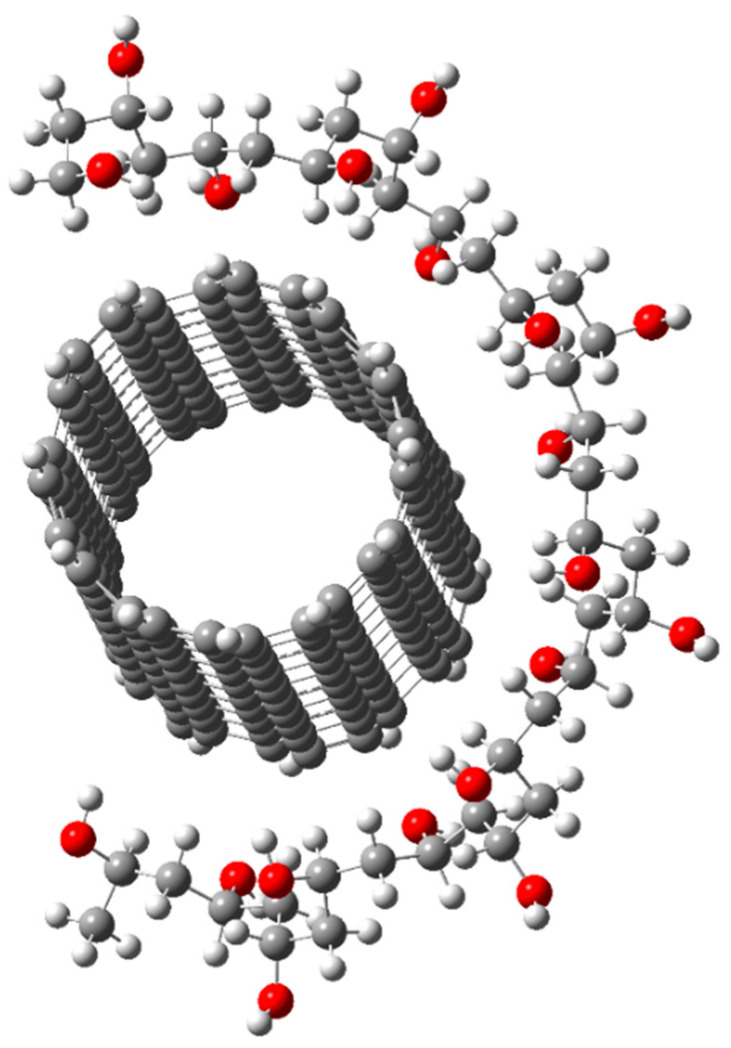
Model of PMMA/CNT system.

## Data Availability

The data are contained within the article.

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
