# Peer review of "Composite Nanomaterials Based on Polymethylmethacrylate Doped with Carbon Nanotubes and Nanoparticles: A Review"

_polymers, 2024, doi:10.3390/polym16091242_

Round 1
Reviewer 1 Report
Comments and Suggestions for Authors
Reviewer comments
Manuscript ID: Polymers-2966612
Title: Composite Nanomaterials Based on Polymethylmethacrylate Doped with Carbon Nanotubes and Nanoparticles: A Review
Journal: Polymers
The manuscript under review provides a comprehensive review of composite nanomaterials composed of polymethylmethacrylate (PMMA) doped with carbon nanotubes (CNTs) and nanoparticles.
The review covers various aspects related to the synthesis, properties, and potential applications of these composite materials. While the paper is well-aligned with the thematic scope of the journal, there are areas that require improvement. Therefore, I suggest the following revisions to enhance the quality and depth of the manuscript:
1- Line 18: rewrite this sentence “ The results of experimental studies of the effect of the number of CNTs on the mechanical and ...”
2- Line 138: correct the references numbers “[30-34]” and not “[20-24]”
3- Line 341: correct the reference number “[75]” and not “[65]”
4- Line 463: change “…and, but also” by “…but also”
5- Authors are encouraged to incorporate a comparative analysis of various types of PMMA-based nanocomposites, along with their respective performance characteristics, into the manuscript.
6- Additionally, providing further results could enrich the conclusion by offering insights into the relative advantages and limitations of each type. Addressing these suggested improvements could enhance the clarity, relevance, and impact of the conclusion.
7- Typos and grammar problems need to be corrected properly, authors should carefully check through the manuscript before submitting a revision
The reviewer recommends that the author do minor revision to the manuscript.
Comments on the Quality of English LanguageThe reviewer would recommend that the authors proofread the article thoroughly for typos and grammatical errors.
Reviewer 2 Report
Comments and Suggestions for Authors
The authors have reviewed several properties and applications of PMMA nanocomposites, with a special emphasize put on carbon nanotubes as the filler. Whereas the paper is interesting, there are several weak points that the authors should consider.
1) Figs. 1 and 2 can be dropped. It is very likely that the readers of the paper know what carbon nanotubes are and how the structure of the nanotubes wall can be constructed from a graphene plane. similarly, the text containing lines 184 to 200 can be shortened.
2) The word "tubulene" is not used anymore. The authors probably use this word for single-walled carbon nanotubes. They may refer to them by the acronym SWCNT.
3) The text containing lines 274 to 292 is strange. Most of the sentence are duplicated. Example "Composites were obtained using hot pressing and injection molding methods" in line 274 is identical to the sentence starting end of line 2B2. Please clean the text.
4) Same remarks for lines 342 to 346.
5) It is interesting to discover that PMMA nanocomposites have applications in dentures (bottom of page 3, bottom of page 6). Perhaps the authors could be a little more precise there (an in other places as well). What part of dentures are bases on PMMA? In case of false teeth, how is the composite made white? Usually, polymer doped with graphite or nanotubes are black.
6) In Section 2.4, the word "shield" is better than "screen" in the context of electromagnetic interference
7) Please define the acronym OUNT (line 388)
8) The sentence in lines 406 to 408 is difficult to understand. Please, correct.
9) The details contained in lines 446 to 450 should not appear in a review paper. The interested reader can obtain all the details in the cited references. Same remark for line 404 where it is not necessary to mention the B3LYP hybrid potential and the Gaussian basis used.Why do the authors justify the importance of Fig. 5, unless it has not been published elsewhere?
10) Please, add the publication year of references 13, 14, 45, 46, 56, 77
Some sentences are difficult to understand. See also my report.
Round 2
Reviewer 2 Report
Comments and Suggestions for Authors
I am happy with the revised version of this review paper. The authors have followed the recommendations contained in my report. It seems also that interesting data have been added by the authors in response to an other referee. The paper can be published in Polymers as it is without further delay.
Comments on the Quality of English LanguageThe paper is understanding as it is. A few English constructions should be corrected (eg, "arm-chair" should be written as "armchair).